# Components of the Complement Cascade Differ in Polycystic Ovary Syndrome

**DOI:** 10.3390/ijms232012232

**Published:** 2022-10-13

**Authors:** Alexandra E. Butler, Abu Saleh Md Moin, Thozhukat Sathyapalan, Stephen L. Atkin

**Affiliations:** 1Research Department, Royal College of Surgeons in Ireland Bahrain, Adliya 15503, Bahrain; 2Academic Endocrinology, Diabetes and Metabolism, Hull York Medical School, Hull HU67RU, UK

**Keywords:** polycystic ovary syndrome, complement factors, C3, properdin, factor B

## Abstract

Complement pathway proteins are reported to be increased in polycystic ovary syndrome (PCOS) and may be affected by obesity and insulin resistance. To investigate this, a proteomic analysis of the complement system was undertaken, including inhibitory proteins. In this cohort study, plasma was collected from 234 women (137 with PCOS and 97 controls). SOMALogic proteomic analysis was undertaken for the following complement system proteins: C1q, C1r, C2, C3, C3a, iC3b, C3b, C3d, C3adesArg, C4, C4a, C4b, C5, C5a, C5b-6 complex, C8, properdin, factor B, factor D, factor H, factor I, mannose-binding protein C (MBL), complement decay-accelerating factor (DAF) and complement factor H-related protein 5 (CFHR5). The alternative pathway of the complement system was primarily overexpressed in PCOS, with increased C3 (*p* < 0.05), properdin and factor B (*p* < 0.01). In addition, inhibition of this pathway was also seen in PCOS, with an increase in CFHR5, factor H and factor I (*p* < 0.01). Downstream complement factors iC3b and C3d, associated with an enhanced B cell response, and C5a, associated with an inflammatory cytokine release, were increased (*p* < 0.01). Hyperandrogenemia correlated positively with properdin and iC3b, whilst insulin resistance (HOMA-IR) correlated with iC3b and factor H (*p* < 0.05) in PCOS. BMI correlated positively with C3d, factor B, factor D, factor I, CFHR5 and C5a (*p* < 0.05). This comprehensive evaluation of the complement system in PCOS revealed the upregulation of components of the complement system, which appears to be offset by the concurrent upregulation of its inhibitors, with these changes accounted for in part by BMI, hyperandrogenemia and insulin resistance.

## 1. Introduction

Polycystic ovary syndrome (PCOS) is the most common endocrine disorder in premenopausal women and leads to an increased prevalence of type 2 diabetes, hypertension and, potentially, cardiovascular disease [1], which may be mediated through inflammation, although the underlying pathophysiology remains unclear [2]. Elevated complement pathway proteins for both the classical and alternate cascades, including C3, C4, properdin, factor B and factor D [3], have been reported in PCOS (Figure 1), although their expression was correlated to obesity and insulin resistance [3]. Complement protein studies in PCOS have been conflicting, with one confirmatory study reporting that C3 may be elevated and related to inflammation [4], whilst others report that C3 levels are unchanged [5]. In an exercise study, C3 was reported to be higher in PCOS compared to controls at baseline and, following 3 months of supervised exercise, there was a reduction in levels of complement factors C1q, C3, C4, factor B, factor H and properdin, but only in controls [6].

The complement system has three major pathways to activation, classical, alternate and lectin, and is the chief component of innate immunity that not only participates in inflammation, but also acts to enhance the adaptive immune response [7]. Simplistically, the classical pathway is activated through C4, the alternative pathway through C3 and the lectin pathway through C2, although all arms lead to a common pathway as shown in Figure 1. Complement activation is involved in several disease processes, including diabetes [8] and cardiovascular disease [9], both of which are associated with PCOS. Furthermore, there is increasing evidence that PCOS is associated with an increased risk from infection and its sequelae [10]. 

Few studies have looked at the complement system in PCOS, and none in a comprehensive manner; therefore, this study was undertaken in a PCOS population versus age-matched controls to determine whether the complement factor proteins were independently associated with PCOS. 

## 2. Results

As can be seen in Table 1, the two cohorts were matched for age, but those with PCOS had a greater BMI, and showed increased insulin resistance, hyperandrogenemia and increased CRP (as a marker of inflammation). 

The results of the complement factors are shown in Table 2 for the PCOS and control subjects. 

### 2.1. Alternative Pathway (AP) Proteins of Complement Activation in PCOS

The levels of the alternative pathway (AP) protein of complement activation component C3 were higher in PCOS; however, the functional fragments of the C3 cleavage, C3a and C3b, were not different in PCOS compared to non-PCOS women. Among the positive regulators of the AP, factor B (F-B) was higher in PCOS, but factor D (F-D) did not differ in PCOS compared to the control. The levels of properdin were higher in PCOS. Among the negative regulators of the AP, the levels of factor H (F-H) and factor I (F-I) were higher in PCOS. Complement factor H-related protein (CFHR5) levels were higher in PCOS. The levels of the degradation products (by F-I) C3b, iC3b and C3d were also higher in PCOS. 

### 2.2. Lectin Pathway (LP) Proteins of Complement Activation in PCOS

The levels of C2, the component of the lectin pathway of complement activation, were higher in PCOS compared to the controls. The levels of mannose-binding lectin (MBL) were not different between PCOS and the control.

### 2.3. Classical Pathway (CP) Proteins of Complement Activation in PCOS

There was no change in classical pathway component C1q and C1r levels between PCOS and the control. 

C4a and C4b did not differ between groups, nor were there any differences in C3adesArg, C5, C5b-6 complex or C8 (Table 2). 

### 2.4. Correlations of Complement Activation-Related Proteins with Hyperandrogenemia, Insulin Resistance and BMI

Hyperandrogenemia correlated positively with properdin and iC3b in PCOS, whilst C1r correlated negatively in controls (Figure 2). 

In view of the apparent correlations of the proteins with insulin resistance and BMI, seen in Figure 2 and Figure 3, a multivariable regression model was undertaken, adjusting the protein results for BMI and insulin levels to determine if the adjusted PCOS versus the control differed: C3b, iC3b, C3d, factor I, CFHR5, factor B, factor H and C5a all remained significantly different (*p* < 0.05) after accounting for BMI and HOMA. This may also be visually seen in Appendix A, where it is apparent that the matched obese controls versus PCOS demonstrated that the complement pathway protein levels were significantly different between those groups.

Insulin resistance (HOMA-IR) positively correlated with iC3b and factor H in PCOS and controls; additionally, in controls, properdin, C3a, C3d, factor I, factor B, C5a, C5b,6 complex and C5 all positively correlated with HOMA-IR (Figure 2).

A correlation with BMI showed positive associations with C3d, factor I, factor D, CFHR5, factor B, factor H and C5a in PCOS (Figure 3). In controls, positive correlations with BMI were found for C3d, factor I, factor B, factor H, C5a, iC3b, C3, C3adesArg, C3a, C4 and C5 (Figure 3 and Figure 4).

In view of the apparent alternative pathway activation, the von Willebrand factor was measured, as this has been shown to be an important modulator in the pathway and was shown to be elevated in PCOS (*p* = 0.04) (Table 2).

## 3. Discussion

Here, we report the complement activation system and the correlations with hormonal and metabolic parameters in women with PCOS. The current data demonstrated that the alternative pathway proteins of the complement system were primarily overexpressed in PCOS, with increased C3, properdin and factor B; however, this appeared to be balanced by an increase in the inhibitory factors of this pathway, CFHR5, factor H and factor I, suggesting an inefficient complement activation system in the normal fasting condition in PCOS; however, the findings could be accounted for in part by BMI, hyperandrogenemia and insulin resistance. Our data indicated that there was no difference in key recognition molecules of the classical pathway (C1r, C1q) and lectin pathway (MBL) of complement activations in PCOS subjects.

Target recognition and initiation are two key steps of complement activation, and through which, each pathway is triggered. Our current data suggest that the pattern recognition molecules for both the classical pathway (CP) (C1q and C1r) and the lectin pathway (LP) (MBL) of complement activation were not different in PCOS, suggesting that, in the resting physiological condition, classical and lectin pathways of complement activation are not activated in PCOS. The underlying mechanism for this phenomenon might involve low levels of circulating IgG in PCOS, as classical pathway components recognize IgG- and IgM-containing immune complexes for activation. Low levels of the main subclasses of IgG and their negative correlation with cortisol and ACTH have been reported in non-obese subjects with PCOS [11]. Previous research also demonstrated that testosterone treatment reduced IgG production by 59.0% and that of IgM by 61.3% compared with controls [12]; this further supports our data where a negative correlation between testosterone levels and C1r was observed in PCOS, but not in control women, suggesting testosterone-mediated inhibition of the classical pathway of complement activation in the normal/resting physiological condition in PCOS. 

The LP pattern recognition molecule MBL is a soluble protein produced mainly by hepatocytes. A previous report demonstrated that MBL synthesis in humans was influenced by growth hormone [13]; however, growth hormone secretion is impaired and related to hyperandrogenism in non-obese patients with PCOS [14,15]. Therefore, it is highly likely that growth hormone plays a role in the reduced level of MBL that, in turn, leads to inefficient pattern recognition of LP complement activation in PCOS. 

### 3.1. Alternative Pathway Modulation in PCOS

In normal physiological conditions, the alternative pathway (AP) is the active and dominant complement pathway. The AP scans for invasive pathogens (or eliminates dying host cells) by being constitutively active at a low level. C3 is the common central component of the complement system and each of the three pathways results in the cleavage of the inactive C3 protein into C3a and C3b, the functional fragments. C3a acts as an inflammatory mediator and C3b is an opsonin, which can bind covalently and tag surfaces (of either foreign pathogen or host cell) in the area close to where it is generated (reviewed in [16]). Our current data demonstrated elevated levels of C3 in PCOS; however, the functional fragments of the C3 cleavage (C3a and C3b) were not different in PCOS compared to the control, suggesting the reduced activity of the C3 cleaving enzyme, alternative pathway C3-convertase (C3bBb). Serum levels of C3 (including fasting C3 levels) are closely related to adiposity, insulin action and fasting insulin levels, and correlate positively with BMI, and visceral, subcutaneous and total fat volume [17,18]. Moreover, fat ingestion and circulating lipids have also been postulated as possible indicators of serum C3 levels. However, in a study focused on an elderly population, it was demonstrated that serum C3 levels are strongly associated with insulin resistance after adjustment for obesity [19]. In our study, a positive correlation of C3 levels with BMI was only observed in control subjects, not in subjects with PCOS, suggesting C3 levels in PCOS might be regulated by a mechanism independent of obesity. 

### 3.2. Factor B and Factor D in the Alternative Pathway in PCOS

In the AP, two forms of C3 convertases are generated; one is “fluid phase C3 convertase”, C3(H_2_O)Bb, the other being membrane-bound C3 convertase, C3bBb. Circulating C3 undergoes “spontaneous hydrolysis” to form C3(H_2_O) and, with the help of two positive regulators of the AP, factor B (F-B) and factor D (F-D), surface-bound C3 convertase (C3bBb) is formed. Our current data indicate that the level of F-B is increased in PCOS; however, there was no change in the level of F-D, again suggesting that not all the components of the AP are sufficiently available for AP activation in normal physiological conditions in PCOS. Consistent with the fact that F-B plays a crucial role in the latter stages of adipocyte differentiation and lipid droplet formation [20], F-B positively correlated with BMI both in the control and in PCOS, indicating that in PCOS, F-B may be involved in adipocyte differentiation (when inadequate F-D is available for AP activation). 

### 3.3. Factor H in the Alternative Pathway in PCOS

Factor H (F-H), the soluble inhibitor of the C3 convertase, competes with F-B to bind to C3b to regulate the AP and the amplification loop of the complement pathway [21]. Factor I (F-I), another inhibitor of the AP, is a serine protease (SP) that cleaves C3b in the presence of different molecules, including F-H. The protease activity of F-I leads to the generation of the degradation product of C3b, iC3b, which is unable to bind F-B [22]. Consistent with a previous report [3], we also found higher F-H levels in PCOS and a strong association of F-H with BMI both in controls and PCOS. In addition, our data demonstrated that fasting F-I was higher and correlated with BMI in PCOS. Therefore, elevation of both inhibitors in PCOS likely reflects dysregulation of the AP in PCOS. 

This concept is further supported by complement factor H-related protein 5 (CFHR5) levels in PCOS. CFHR5 is a 65-kDa plasma glycoprotein produced in the liver and has been reported to have complement regulatory activities. Like most other CFHRs, CFHR5 binds to heparin and C3b on self surfaces and acts as a F-I cofactor in C3b cleavage [23]. CFHR5 and F-H compete to bind to C3b; CFHR5 also binds to iC3b, a product of C3b cleavage [24]. We found higher fasting CFHR5 levels and a strong positive correlation with BMI in PCOS, further supporting the hyperactivation of regulatory proteins in the AP during fasting in PCOS. 

Several proteins have been shown to enhance F-I-mediated cleavage in the presence of cofactors. For example, the von Willebrand factor (vWF) has been reported to enhance the efficacy of F-H as a cofactor of F-I [25], as well as through direct cofactor activity [26]. vWF levels were significantly higher in PCOS compared to controls, suggesting an enhanced F-I activity in PCOS and, thus, dysregulation of the AP in PCOS. Moreover, the acute-phase pentraxin-family protein, C-reactive protein (CRP), has been suggested to inhibit the activation of the alternative pathway (AP) [27]. CRP has been observed to bind the alternative complement pathway inhibitor factor H (F-H) to potentially recruit it to the areas of tissue damage, hence, limiting AP activation [28]. The levels of CRP were significantly higher in PCOS in our study, strongly suggesting CRP-mediated dysregulation of AP in PCOS.

### 3.4. Properdin in the Alternative Pathway in PCOS

The AP C3 convertase (C3bBb) is a short-lived complement complex, having an ~90 s half-life [29]; therefore, complex stabilization is necessary for host defense. Properdin, an important positive regulator of the complement system, stabilizes the C3 convertase [30] by interacting with both the C345C domain of C3b and the VWF-A domain of Bb [31]. Our data indicated higher properdin levels in PCOS; however, since the levels of C3b were not different in PCOS, it is tempting to speculate that properdin was unable to stabilize the AP C3 convertase C3bBb in PCOS. A previous report demonstrated that the levels of C3 and properdin were reduced in women versus men; however, we found both C3 and properdin were elevated in PCOS, suggesting an androgen-mediated elevation of these two proteins in PCOS. Properdin is secreted by monocytes, macrophages and T lymphocytes; androgen receptor signaling positively regulates monocyte development [32]. A positive correlation between testosterone and properdin levels was also observed in our study. Taken together, the elevated properdin levels may not be reflective of AP activation but, rather, may be indicative of testosterone-mediated elevation of properdin levels in PCOS. 

### 3.5. Complement Factor C3 in PCOS

Complement factor C3 has been studied, particularly in the setting of obesity-induced insulin resistance because of its cleavage product, C3adesArg, which is also termed acylation-stimulating protein (ASP). In the AP, C3 activation generates C3a, which is quickly converted to C3desArg by plasma carboxypeptidase N [33]. In humans, ASP levels were found to be elevated in obese versus non-obese subjects and their levels decreased in obese subjects who experienced diet- or fasting-induced weight loss [34,35]. Consistent with previous studies, a positive correlation of C3adesArg with BMI was observed in control subjects; however, no such correlation was found in patients with PCOS, suggesting an altered C3adesArg response in PCOS. Moreover, like C3a, fasting C3adesArg levels did not differ in PCOS (despite increased fasting C3 levels in PCOS), which further supports the concept of dysregulated C3 convertase activity in PCOS.

### 3.6. Complement Factor C5 in PCOS

In the final stages of the complement cascade, the C5 convertase cleaves C5 (an inert molecule) into C5a (a potent anaphylatoxin) and C5b (a bioactive fragment). C5b instigates assembly of the later component proteins (C6, C7, C8 and multiple copies of C9) and causes their insertion into the cell membrane to create the membrane attack complex C5b-9 (MAC), which ultimately forms a 10 nm-wide pore in the target membrane [36]. Fasting levels of C5 and C5b,6 complex did not differ in PCOS; however, C5a levels were higher in PCOS. This suggests the unavailability of the initiating components (C5b, which, after cleavage, experiences conformational change facilitating interaction with C6) of MAC formation in the fasting state of PCOS. Further studies may be required to explain the increased C5a levels in PCOS; however, emerging evidence suggests that, aside from the three well-recognized pathways, additional complement activation pathways are present in plasma [37]. In addition, it has previously been demonstrated that coagulation/fibrinolysis proteases (thrombin, human coagulation factors, IX, XI, X and plasmin) could be natural C3 and C5 convertases, able to generate anaphylatoxins that are biologically active [38]. In our study, we found that among the above coagulation factors, factor IX was significantly higher in PCOS, potentially suggesting a role for factor IX in the elevation of fasting C5a in patients with PCOS.

### 3.7. Complement Homeostasis Dysregulation in PCOS

Whilst it appears that there is dysregulated complement activation and the system is in homeostasis on a daily basis, hypothetically, if there were a perturbation in the system due to an intervening condition, then this may lead to an exaggerated response [39]. Of note, this may be the case for those women with PCOS who contract COVID-19 disease; PCOS subjects appear at higher risk for severe COVID-19 infection [10,40] and it is well known that complement activation in COVID-19 is associated with severe respiratory symptoms [41].

Overall, it can be difficult to determine the relative contributions of obesity, hyperandrogenemia and insulin resistance in the inherent dysfunction of PCOS [42,43]. Notably, the underlying disease process of PCOS may also be affected by ethnicity [44]. The complement protein results reported here are in accord with some studies on PCOS [3,4] but not others [5]; however, those studies had not taken into account the underlying pathophysiology of obesity, insulin resistance and inflammation that are addressed here.

Limitations of this study include that, whilst age did not differ between the two groups, the PCOS group had a higher BMI. In addition, all the subjects were white, and the effect of ethnicity is unknown. Further, the SOMAscan platform reports the results as relative fluorescent units (RFU) and it is not possible to convert RFUs to protein concentration. It was not possible to undertake any functional studies to confirm the hypothesis that, firstly, the alternative complement system was in balance and secondly that, if this balance was disrupted, then an enhanced complement response may result in the setting of PCOS. It would also be important for future studies to address the role of the PCOS phenotype [45] in the presence and absence of obesity, ideally with cohorts matched for insulin resistance.

## 4. Materials and Methods

We determined plasma classical and alternative complement pathway protein levels in PCOS (*n* = 137) and control (*n* = 97) women following their recruitment to a PCOS biobank (ISRCTN70196169) from 2012–2017. All patients gave written informed consent. Samples plus clinical data were acquired from the UK PCOS Genetic Biobank, with approval from the Newcastle and North Tyneside Ethics Committee (reference number 10/H0906/17). Baseline data for the 137 PCOS patients and 97 controls are shown in Table 1. 

As detailed previously [46], “all women were Caucasian. The diagnosis of PCOS was based on at least two out of three of the diagnostic criteria of the Rotterdam consensus as detailed previously [46]; namely clinical and biochemical evidence of hyperandrogenism (Ferriman-Gallwey score >8; free androgen index >4, total testosterone >1.5 nmol/L), oligomenorrhea or amenorrhoea and polycystic ovaries on transvaginal ultrasound. Nonclassical 21-hydroxylase deficiency, hyperprolactinemia, Cushing’s disease and androgen secreting tumors were excluded by appropriate tests. The baseline study measurements have been described in detail previously [47] and the demographic data for the PCOS and control women is shown in Table 1. All the control women recruited by advert had regular periods, no clinical or biochemical hyperandrogenism, no polycystic ovaries on ultrasound, no significant background medical history and none of them were on any medications including oral contraceptive pills or over-the-counter medications. 

Fasting bloods were centrifuged at 3500× *g* for 15 min, placed into aliquots and frozen at −80 °C until analysis. The bloods were analysed for sex hormone binding globulin (SHBG), insulin (DPC Immulite 200 analyser, Euro/DPC, Llanberis, UK), and plasma glucose (Synchron LX20 analyser, Beckman-Coulter, High Wycombe, UK). Free androgen index (FAI) was calculated by dividing the total testosterone by SHBG, and then multiplying by one hundred. Insulin resistance (IR) was calculated using the homeostasis model assessment (HOMA-IR). Serum testosterone was quantified using isotope-dilution liquid chromatography tandem mass spectrometry (LC-MS/MS) [48]. 

Circulating levels of complement pathway proteins were determined by Slow Off-rate Modified Aptamer (SOMA)-scan plasma protein measurement, the details of which have been previously reported [49]. Normalization of raw intensities, hybridization, median signal and calibration signal were performed based on the standard samples included on each plate, as previously described [50].

Version 3.1 of the SOMAscan Assay targeted those proteins specifically involved in the classical and alternative complement pathways in the SOMAscan panel, some of which were reported by others previously [3]. These included the following complement system proteins for the classical, alternative and lectin pathways: C1q, C1r, C2, C3, C3a, iC3b, C3b, C3d, C3adesArg, C4, C4a, C4b, C5, C5a, C5b-6 complex, C8, properdin, factor B, factor D, factor H, factor I, mannose-binding protein C (MBL), complement decay-accelerating factor (DAF) and complement factor H-related protein 5 (CFHR5) (Figure 1, Table 2).

### Statistics

A power analysis (nQuery version 9, Statsols, San Diego, CA, USA) was undertaken for the C3 protein previously reported to be different in PCOS [3]. For 80% power and an alpha of 0.05 with a common standard deviation (SD) of 0.37, the number of subjects required was 23. Trends in the data were visually inspected and statistically evaluated for normality. Student’s *t*-tests were applied on Gaussian-distributed data, whilst Mann–Whitney (non-parametric) tests were applied on non-Gaussian data as determined by the Kolmogorov–Smirnov test. All analyses were performed using R version 4.0.0 (R Foundation for Statistical Computing, Vienna, Austria. https://www.R-project.org/, accessed on 4 July 2022).

## 5. Conclusions 

In conclusion, this comprehensive evaluation of the complement system in PCOS revealed apparent dysregulation of the alternative complement system that appears to be offset by the concurrent upregulation of its inhibitors, and with the changes in part accounted for by BMI, hyperandrogenemia and insulin resistance.

## Figures and Tables

**Figure 1 ijms-23-12232-f001:**
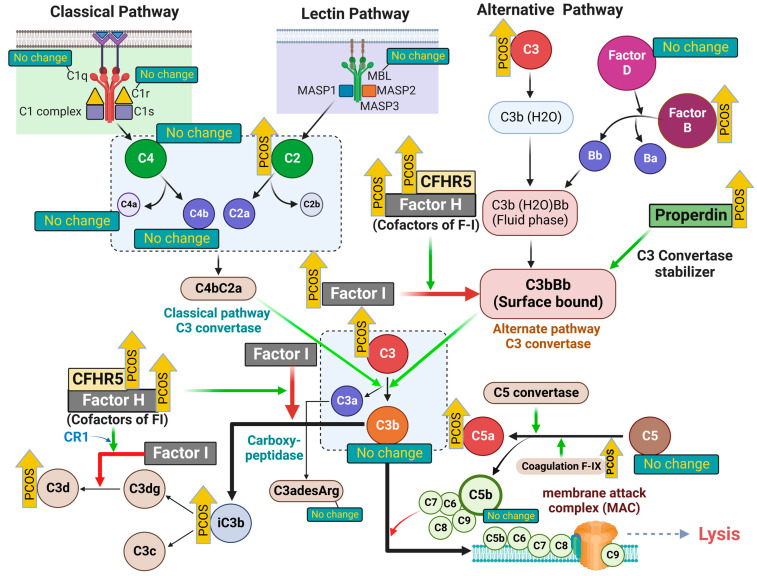
A schematic illustrating the initiating proteins of the classical, alternative and lectin complement cascades. Upward yellow arrows (labeled “PCOS”) indicate the proteins that were increased in women with PCOS. The green arrows in the illustration indicate the enzymatic activity or positive regulation, whereas the red arrows indicate the inhibition of pathways. Proteins that are not different in the PCOS cases are also indicated accordingly in the illustration. The illustration was created using BioRender.com (with publication license).

**Figure 2 ijms-23-12232-f002:**
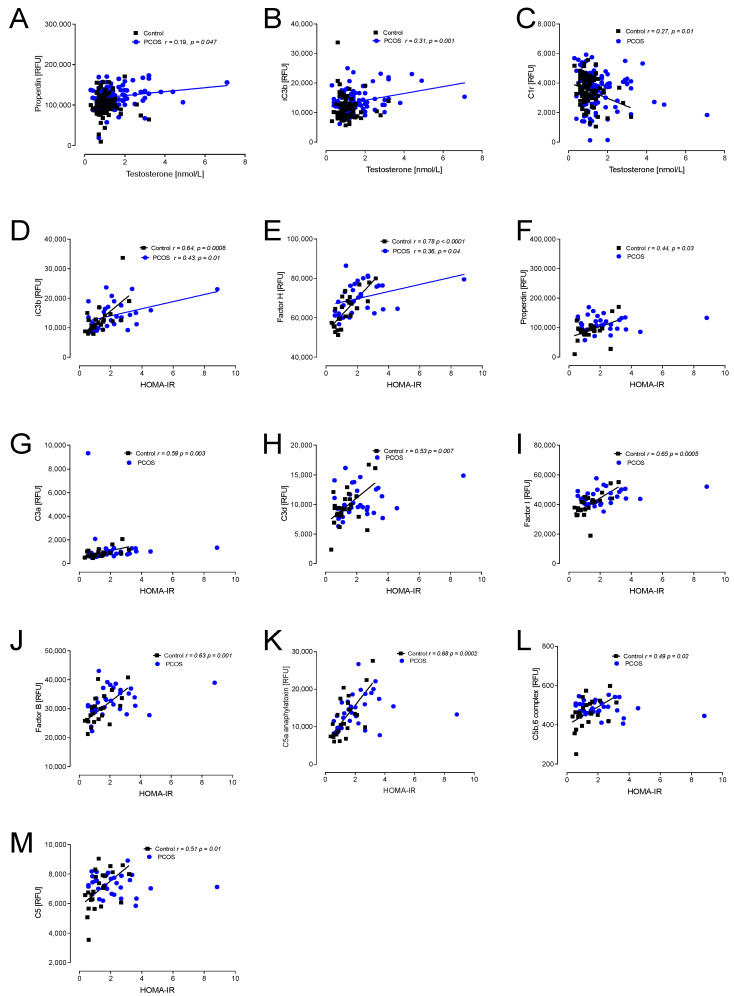
Correlations of complement pathway proteins with hyperandrogenemia (testosterone) and with insulin resistance (HOMA-IR) in patients with PCOS (*n* = 137) and controls (*n* = 97). Proteins that correlated, all positively, with testosterone were properdin (**A**), iC3b (**B**) and C1r (**C**). Proteins that correlated, all positively, with HOMA-IR were iC3b (**D**), factor H (**E**), properdin (**F**), C3a (**G**), C3d (**H**), factor I (**I**), factor B (**J**), C5a (**K**), C5b,6 complex (**L**) and C5 (**M**). RFU—relative fluorescent units.

**Figure 3 ijms-23-12232-f003:**
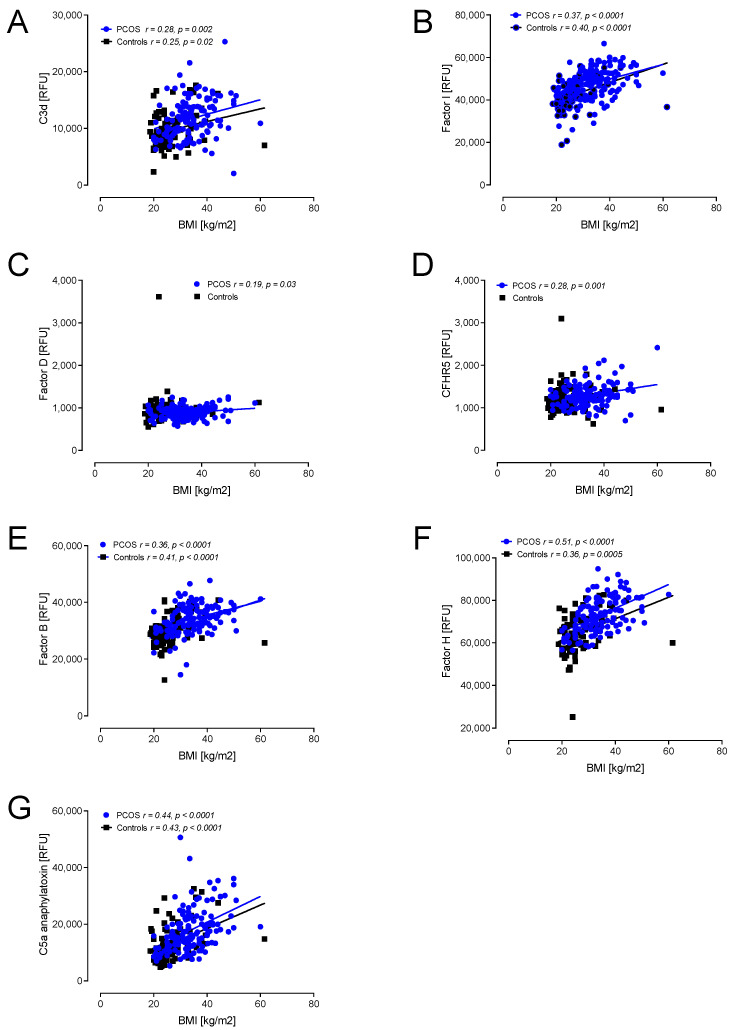
Correlations of complement pathway proteins with body mass index (BMI) in patients with PCOS (*n* = 137) and controls (*n* = 97). Proteins that correlated, all positively, with BMI were C3d (**A**), factor I (**B**), factor D (**C**), CFHR5 (**D**), factor B (**E**), factor H (**F**) and C5a (**G**). RFU—relative fluorescent units.

**Figure 4 ijms-23-12232-f004:**
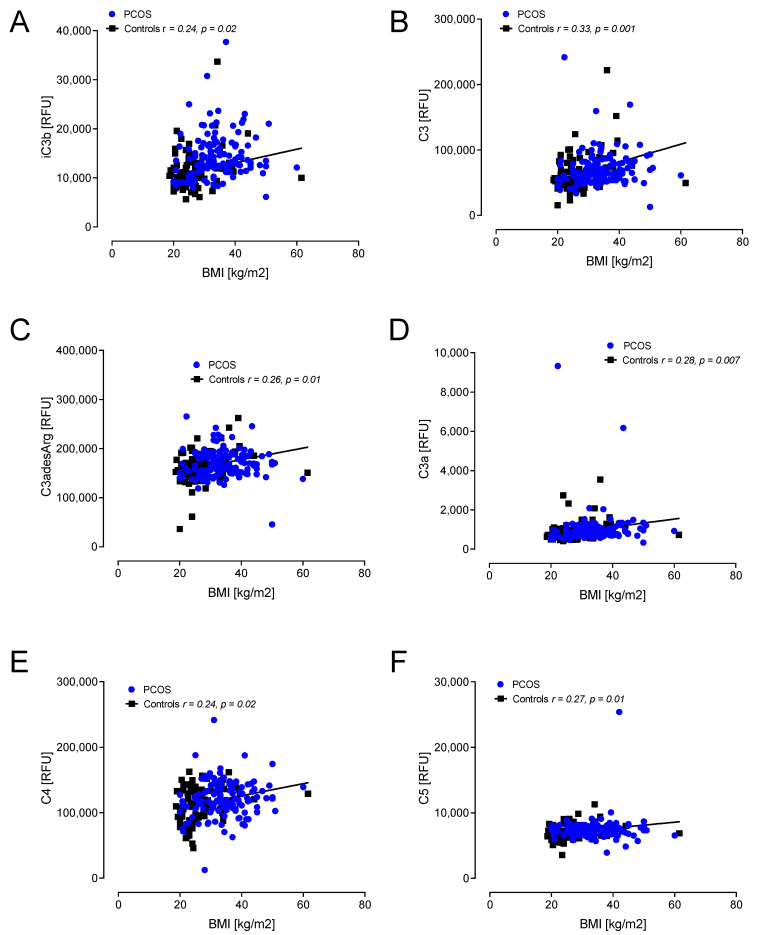
Correlations of complement pathway proteins with body mass index (BMI) in control women (*n* = 97) only. relative fluorescent units (RFU). Proteins that correlated, all positively, with BMI in control women only were iC3b (**A**), C3 (**B**), C3adesArg (**C**), C3a (**D**), C4 (**E**) and C5 (**F**). RFU—relative fluorescent units.

**Table 1 ijms-23-12232-t001:** Demographics and baseline hormonal and metabolic parameters of the polycystic ovary syndrome (PCOS) subjects and controls.

Baseline Demographics	PCOS (*n* = 137)	Controls (*n* = 97)	*p*-Value
	Mean (SD)	Mean (SD)	
Age (years)	29.1 (6.1)	29.6 (6.5)	0.09
BMI (Kg/m²)	34.1 (7.5)	26.7 (6.6)	<0.0001
Weight (Kg)	96.5 (23.7)	74.4 (18.4)	<0.0001
Insulin (IU/mL)	10.2 (6.1)	6.2 (3.2)	0.001
HOMA-IR	3.8 (0.6)	1.6 (0.2)	<0.005
CRP (mg/L)	4.4 (4.2)	2.4 (3.9)	0.0008
SHBG (nmol/L)	42.5 (39.6)	77.5 (78.4)	0.0003
Testosterone (nmol/L)	1.6 (1.0)	1.05 (0.48)	<0.0001

BMI—body mass index; HOMA-IR—homeostasis model of assessment-insulin resistance; CRP—C-reactive protein; SHBG—sex-hormone-binding globulin.

**Table 2 ijms-23-12232-t002:** Complement cascade and related proteins in patients with polycystic ovary syndrome (PCOS) fulfilling all three diagnostic criteria versus controls. Data presented as mean ± 1 standard deviation of relative fluorescent units (RFU).

	PCOS	Control	*p*-Value
Properdin	119,125 (26,794)	102,491 (26,374)	0.001
C3b	110,730 (68,743)	105,010 (59,972)	0.5
iC3b	144,776 (4563)	11,445 (4188)	0.001
C3	71,028 (25,536)	63,896 (26,822)	0.04
C3adesArg	168,929 (28,060)	163,048 (30,262)	0.12
C3a	1045 (870)	863 (465)	0.06
C3d	11,905 (3645)	9816 (2890)	0.001
C4	121,193 (26,829)	114,603 (25,411)	0.06
C4a	116,060 (4320)	116,497 (3708)	0.42
Factor I	47,519 (6883)	42,304 (6958)	0.001
Factor D	898 (146)	933 (314)	0.24
C2	2694 (386)	2540 (378)	0.002
Complement factor H-related 5	1298 (264)	1206 (303)	0.01
Factor B	34,162 (5281)	30,078 (5054)	0.002
Factor H	72,843 (8954)	64,750 (9119)	0.001
C5a	17,983 (7875)	13,106 (6113)	0.001
C5b, 6 complex	490 (51)	493 (194)	0.87
C5	7432 (1692)	7071 (1088)	0.06
C1q	38,519 (7784)	37,027 (7612)	0.14
C1r	3414 (1185)	3497 (967)	0.57
C4b	846 (913)	889 (632)	0.69
C8	2403 (539)	2337 (489)	0.34
MBL	16,835 (7427)	16,298 (7912)	0.59
DAF	13,958 (2717)	14,424 (2323)	0.17
IgG	195,643 (38,449)	207,990 (40,148)	0.02
Thrombin	1154.0 (322.0)	967.03 (270.1)	0.68
Plasmin	636.6 (19.0)	619.04 (23.7)	0.56
Coagulation factor IX	11,125.5 (134.1)	9647.8 (143.5)	<0.0001
Coagulation factor XI	1819.3 (24.7)	1794.5 (28.7)	0.52
Coagulation factor X	5778.5 (73.1)	5617.7 (87.2)	0.16
von Willebrand factor	19,849 (36,829)	13,159 (5854)	0.04

## Data Availability

All the data for this study will be made available upon reasonable request to the corresponding author.

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
