# Peer review of "Components of the Complement Cascade Differ in Polycystic Ovary Syndrome"

_ijms, 2022, doi:10.3390/ijms232012232_

Round 1
Reviewer 1 Report
I find the article submitted for review interesting, and the authors' focus on clotting in the PCOS context is fascinating.
The introduction is an excellent introduction to the subject.
Material and methods
I believe Table 1 should be included in this section because it does not present the results but describes the group.
The results in the Results section are presented correctly. However, I believe that duplications are unnecessary, e.g., the p and r values ​​appear twice in the context of correlation (in the figure and in the text); especially in the text, it is not legible; I suggest either inserting a table with p and r values ​​or basing it only on figures.
Discussion - it is too broad, and it is challenging to get through it; I propose dividing it into subsections.
The conclusions are interesting - but because the topic is generally tricky, it would be helpful to support their results - also, in the text (i.e., I think that the authors should at least write in parentheses why there is the dysregulation of the alternative complement system, where there is compensation - based on their obtained results).
Maybe an explanatory figure/diagram would be helpful here (?)
The manuscript language is correct, but there are stylistic errors like "levels of Properdin" (unnecessarily properdin is capitalized). I recommend reading the text carefully once again.
I have no doubts about the ethical aspects of the study.
Author Response
Reviewer 1
I find the article submitted for review interesting, and the authors' focus on clotting in the PCOS context is fascinating.
The introduction is an excellent introduction to the subject.
Thank you for your very positive comments.
Material and methods
I believe Table 1 should be included in this section because it does not present the results but describes the group.
Thank you, Table 1 has now been added to the Methods section as requested.
The results in the Results section are presented correctly. However, I believe that duplications are unnecessary, e.g., the p and r values ​​appear twice in the context of correlation (in the figure and in the text); especially in the text, it is not legible; I suggest either inserting a table with p and r values ​​or basing it only on figures.
Thank you for your comment, with which we entirely agree. As the protein levels with p values are already listed in Table 2, removing these from the text greatly improves the readability.
For the correlation analyses, we have removed the p and r values from the text, so that they now only appear in the Figures.
Discussion - it is too broad, and it is challenging to get through it; I propose dividing it into subsections.
Thank you for your suggestion that we have now addressed by dividing the Discussion into subsections.
The conclusions are interesting - but because the topic is generally tricky, it would be helpful to support their results - also, in the text (i.e., I think that the authors should at least write in parentheses why there is the dysregulation of the alternative complement system, where there is compensation - based on their obtained results).
Thank you for this comment. We described that the alternative pathway (AP) of complement activation is dysregulated in PCOS. The reasons are as follows:
Firstly, in our study, one of the key negative regulators of the alternative pathway (AP) of complement activation, Factor I (F-I), was elevated. F-I is a serine protease found in the plasma that cleaves C3b in the presence of different cofactor molecules, such as Factor H (F-H), membrane cofactor protein (MCP) and complement receptor 1 (CR1). The protease activity of F-I leads to generation of degradation products (C3b, iC3b) that are unable to bind Factor B (F-B) [J Immunol 1996 Jun 15;156(12):4821-32], the key step of generating AP convertase, suggesting inefficient activation of AP in PCOS.
Secondly, Factor H (F-H) is a potent soluble inhibitor of the AP by serving as cofactor for F-I in conversion of active C3b to inactive iC3b. Our data also demonstrated higher levels of F-H in PCOS, further suggesting dysregulation of the alternative pathway (AP) in PCOS.
Thirdly, several proteins have been shown to enhance F-I mediated cleavage in the presence of cofactors. For example, Von Willebrand factor (vWF) has been reported to enhance the efficacy of F-H as a cofactor of F-I [Blood (2014) 123:121–5.] as well as through direct cofactor activity [Blood (2014) 125(6):1034–7.]. vWF levels were significantly higher in PCOS compared to controls, also suggesting enhanced F-I activity in PCOS and thus dysregulation of the AP in PCOS.
Fourthly, the acute phase pentraxin family protein, C-reactive protein (CRP), has been suggested to inhibit the activation of the alternative pathway (AP) [J. Immunol.127:2089]. CRP has been observed to bind the alternative complement pathway inhibitor, Factor H (F-H), to potentially recruit it to areas of tissue damage, hence, limiting AP activation [J Immunol. (1999) 163:3957–62.]. The levels of CRP were significantly higher in PCOS in our study, strongly suggesting CRP-mediated dysregulation of the AP in PCOS.
We had already mentioned the above facts in the discussion (page 10, under the section “Factor H in the alternative pathway in PCOS”). We have now added the additional information noted above in the same section of the Discussion and this now reads:
“Factor H (F-H), the soluble inhibitor of C3 convertase, competes with F-B for binding to C3b to regulate the AP and the amplification loop of the complement pathway (26). Factor I (F-I), another inhibitor of AP, is a serine protease (SP) that cleaves C3b in the presence of different molecules including F-H. The protease activity of F-I leads to the generation of the degradation product of C3b, iC3b, which is unable to bind F-B (27). Consistent with a previous report (3), we also found higher F-H levels in PCOS and a strong association of F-H with BMI both in controls and PCOS. In addition, our data demonstrated that fasting F-I was higher and correlated with BMI in PCOS. Therefore, elevation of both inhibitors in PCOS likely reflects dysregulation of the AP in PCOS.
This concept is further supported by complement factor H related protein 5 (CFHR5) levels in PCOS. CFHR5 is a 65-kDa plasma glycoprotein produced in the liver and has been reported to have complement regulatory activities. Like most other CFHRs, CFHR5 binds to heparin and C3b on self surfaces and acts as a F-I cofactor in C3b cleavage (28). CFHR5 and F-H compete for binding to C3b; CFHR5 also binds to iC3b, a product of C3b cleavage (29). We found higher fasting CFHR5 levels and a strong positive correlation with BMI in PCOS, further supporting the hyperactivation of regulatory proteins of the AP during fasting in PCOS.
Several proteins have been shown to enhance F-I mediated cleavage in the presence of cofactors. For example, Von Willebrand factor (vWF) has been reported to enhance the efficacy of F-H as a cofactor of F-I (30) as well as through direct cofactor activity (31). vWF levels were significantly higher in PCOS compared to controls, suggesting an enhanced F-I activity in PCOS and thus dysregulation of the AP in PCOS. Moreover, the acute phase pentraxin family protein, C-reactive protein (CRP), has been suggested to inhibit the activation of alternative pathway (AP) (32). CRP has been observed to bind the alternative complement pathway inhibitor factor H (F-H) to potentially recruit it to the areas of tissue damage, hence, limiting AP activation (33). The levels of CRP were significantly higher in PCOS in our study, strongly suggesting CRP-mediated dysregulation of AP in PCOS.”
The manuscript language is correct, but there are stylistic errors like "levels of Properdin" (unnecessarily properdin is capitalized). I recommend reading the text carefully once again.
Thank you, the text has now been read through carefully again and stylistic errors amended.
I have no doubts about the ethical aspects of the study.
Reviewer 2 Report
In this manuscript, the authors demonstrate that complement pathway proteins are increased in polycystic ovary syndrome (PCOS) and were affected by obesity and insulin resistance. Furthermore, the authors finished the proteomic analysis of the complement system and suggested that the alternative pathway of the complement system was primarily overexpressed in PCOS, with increased C3 (p<0.05), properdin and Factor B (p<0.01). In addition, inhibition of this pathway was also seen in PCOS, with increase in CFHR5, Factor H and Factor I (p<0.01). The manuscript is well written; however, some clarifications would be useful.
Manuscript Concerns:
1. On page 3, why label the first table as Table 2?
2. In Figure 1, please label all the complement proteins listed in Table 2, such as C2 were significantly increased in PCOS, and C3b is no change between PCOS and control.
3. In Table 2, please change C4A to C4a.
4. In Table 2, please list all the complement proteins as real concentration levels instead of RFU.
5. In Table 1, BMI and HOMA-IR were significantly increased in the PCOS group compared to the Controls. How can you address those complement protein changes due to PCOS but not obesity or insulin resistance when you didn’t separate the obese and non-obese from the PCOS group?
6. Can you address the importance of this work compared to the previous work? (Lewis, R. D., Narayanaswamy, A. K., Farewell, D., & Rees, D. A. (2021). Complement activation in polycystic ovary syndrome occurs in the postprandial and fasted state and is influenced by obesity and insulin sensitivity. Clinical Endocrinology, 94(1), 74-84.)
Author Response
Reviewer 2
In this manuscript, the authors demonstrate that complement pathway proteins are increased in polycystic ovary syndrome (PCOS) and were affected by obesity and insulin resistance. Furthermore, the authors finished the proteomic analysis of the complement system and suggested that the alternative pathway of the complement system was primarily overexpressed in PCOS, with increased C3 (p<0.05), properdin and Factor B (p<0.01). In addition, inhibition of this pathway was also seen in PCOS, with increase in CFHR5, Factor H and Factor I (p<0.01). The manuscript is well written; however, some clarifications would be useful.
Manuscript Concerns:
- On page 3, why label the first table as Table 2?
Thank you, the order of the Tables has been corrected.
- InFigure 1, please label all the complement proteins listed in Table 2, such as C2 were significantly increased in PCOS, and C3b is no change between PCOS and control.
Thank you, this has now been done.
- In Table 2, please change C4A to C4a.
Thank you, this has now been done.
- In Table 2, please list all the complement proteins as real concentration levels instead of RFU.
Thank you for this comment. Unfortunately, whilst it would be useful to list real concentration levels, it is not possible to convert RFU to real concentration using the Soma platform. This has been addressed as a limitation in the Discussion
- In Table 1, BMI and HOMA-IR were significantly increased in the PCOS group compared to the Controls. How can you address those complement protein changes due to PCOS but not obesity or insulin resistance when you didn’t separate the obese and non-obese from the PCOS group?
Thank you for your important comment. You are correct that, if there are differences in BMI and insulin between the groups and BMI/Insulin is associated with both the protein and the PCOS/control group, then this would be confounding. To address this, a multivariable regression model was undertaken, adjusting the protein results for BMI and/or insulin level to determine the adjusted PCOS/Control comparison p-value. This is detailed in the results that reads “ In view of the apparent correlations of the proteins with insulin resistance and BMI seen in Figures 2 and 3, a multivariable regression model was undertaken, adjusting the protein results for BMI and insulin levels to determine if the adjusted PCOS versus control differed: C3b, iC3b, C3d, Factor I, CFHR5, Factor B, Factor H and C5a all remained significantly different (p<0.05) after accounting for BMI and HOMA. This may also be visually seen in Supplementary figure 1 where it is apparent that the matched obese normal versus PCOS showed that the values were significantly different between those groups.”
- Can you address the importance of this work compared to the previous work? (Lewis, R. D., Narayanaswamy, A. K., Farewell, D., & Rees, D. A. (2021). Complement activation in polycystic ovary syndrome occurs in the postprandial and fasted state and is influenced by obesity and insulin sensitivity. Clinical Endocrinology, 94(1), 74-84.)
Thank you for raising this important issue.
Our work complements and adds to the significant findings of the Lewis et al paper. The major advance in this manuscript compared to the Lewis et al publication is twofold.
Firstly, here we have measured a wider swathe of proteins involved in the complement cascade, and it is therefore more comprehensive. Lewis et al reported on C4, C3, C3adesArg, Factor-B, Factor-H, properdin, Factor-D, C5a, C5 and the Terminal complement complex (TCC) (n=10). By contrast, we measured the following 24 complement system proteins for the classical, alternative and lectin pathways: C1q, C1r, C2, C3, C3a, iC3b, C3b, C3d, C3adesArg, C4, C4a, C4b, C5, C5a, C5b-6 complex, C8, properdin, Factor B, Factor D, Factor H, Factor I, Mannose-binding protein C (MBL), Complement decay-accelerating factor (DAF) and Complement factor H-related protein 5 (CFHR5), thus giving a much clearer picture of which pathway (alternative pathway) is largely dysregulated in PCOS and which are not (the classical and leptin pathways). We additionally measured 6 proteins (IgG, thrombin, plasmin and the coagulation factors IX, X and XI) that are relevant to the complement system.
Secondly, Lewis et al measured the proteins by a mixture of techniques: C3 and C4 by nephelometry; C5, TCC and Factor-H using in-house ELISA; C5a(desArg), factor D and properdin using commercial assay kits; and it is not mentioned how Factor-B was assayed.
By contrast, we utilized only a single measurement platform, namely the state-of-the-art Somalogic proteomic analysis (Somascan), a highly multiplexed aptamer-based technology, for measurement of all the proteins presented here, thus providing optimal standardization versus multiple assay techniques.
Round 2
Reviewer 2 Report
Agree to accept!